# Design, Simulation, and Experiment of an LTCC-Based Xenon Micro Flow Control Device for an Electric Propulsion System

**Chang-Bin Guan \*, Yan Shen, Zhao-Pu Yao, Zhao-Li Wang, Mei-Jie Zhang, Ke Nan and Huan-Huan Hui**

Advanced Space Propulsion Lab, Beijing Institute of Control Engineering, Beijing 100094, China; shenyan1978@gmail.com (Y.S.); yzp06@sina.com (Z.-P.Y.); wangzhaoli.bionic@gmail.com (Z.-L.W.); zhang-mj15@tsinghua.org.cn (M.-J.Z.); nanke@buaa.edu.cn (K.N.); huihuanhuan@icloud.com (H.-H.H.)
* Correspondence: guanchangbin@163.com; Tel.: +86-010-68112261

**Abstract:** A xenon micro flow control device (XMFCD) is the key component of a xenon feeding system, which controls the required micro flow xenon (μg/s–mg/s) to electric thrusters. Traditional XMFCDs usually have large volume and weight in order to achieve ultra-high fluid resistance and have a long producing cycle and high processing cost. This paper proposes a miniaturized, easy-processing, and inexpensive XMFCD, which is fabricated by low-temperature co-fired ceramic (LTCC) technology. The design of the proposed XMFCD based on complex three-dimensional (3D) microfluidic channels is described, and its fabrication process based on LTCC is illustrated. The microfluidic channels of the fabricated single (9 mm diameter and 1.4 mm thickness) and dual (9 mm diameter and 2.4 mm thickness) XMFCDs were both checked by X-ray, which proved the LTCC method's feasibility. A mathematical model of flow characteristics is established with the help of finite element analysis, and the model is validated by the experimental results of the single and dual XMFCDs. Based on the mathematical model, the influence of the structure parameters (diameter of orifice and width of the groove) on flow characteristics is investigated, which can guide the optimized design of the proposed XMFCD.

**Keywords:** xenon micro flow control device (XMFCD); low-temperature co-fired ceramic (LTCC); xenon feeding system; electric propulsion system; flow characteristic

## 1. Introduction

Compared with the traditional chemical propulsion system, the electric propulsion system has the advantages of higher specific impulse, simple composition, and no pollution [1]. It has broad application prospects in satellite attitude and orbit control, deep space exploration, interplanetary flight, and other fields. The actuator of the electric propulsion system is an electric thruster, whose principle is to ionize the gas working medium and then accelerate the ion to generate the thrust through an external electric field. Xenon is the ideal working medium for electric thrusters due to its high molecular weight, low ionization energy and easy storage [2].

The xenon feeding system is responsible for the storage, pressure regulation, and flow control of the working medium. It is the key subsystem of the electric propulsion system and belongs to the field of high precision fluid control. The xenon micro flow control device (XMFCD) is the most important component of the xenon feeding system because its control accuracy of the micro xenon flow (μg/s–mg/s) determines the thrust accuracy, working efficiency, and service life of the electric thruster [3]. The core problem of the XMFCD is to achieve ultra-high fluid resistance in a limited volume.

Different XMFCDs have been developed through different implementation methods of ultra-high fluid resistance. The OKB Fakel company of Russia used a capillary type XMFCD to control the micro xenon flow to the electric thruster of SMART-1, which is made of a long-winded capillary tube [4]. The Mott Corporation of USA developed a porous metal type MFC, which is sintered from metal powder [5]. This porous metal type XMFCD was successfully applied in Deep Space 1 [6] and Dawn [7] space detectors. Besides, Northwest Institute for Nonferrous Metal Research of China also produced this type of XMFCD and used it in a hall electric propulsion system [8]. Vacco Industries of USA used chemically etched metal disks as XMFCDs, which realized ultra-high fluid resistance by chemically-etched microfluidic channels [9]. Lee Company invented a stacked-disk containing tangentially machined flow passages, which can be used as an XMFCD [10]. The above XMFCDs all have their shortcomings: (1) The capillary type XMFCD needs very long length to achieve ultra-high fluid resistance, which leads to big weight and volume, (2) the porous metal type XMFCD is heavy and easily produces contaminated particles that causes blockage, (3) the chemically-etched type and the stacked disk type of XMFCDs both have high cost and long production cycle because of their special processing and assembly technologies (such as laser drilling, femtosecond processing, lithography processing, and diffusion welding). Besides, all the above XMFCDs are made of metal, which results in heavy weight and difficulty in constructing microfluidic channels. With the great demand for low-cost and light-weight electric propulsion systems for microsatellites and commercial satellites, the above XMFCDs are no longer able to meet the requirements of the xenon feeding system.

Low temperature co-fired ceramics (LTCC) have been used for almost twenty years to produce a multilayer substrate for packaging integrated circuits [11]. Recently, the multi-layer approach has also been applied to complex three-dimensional (3D) microfluidic structures used as platforms for the fabrication of miniaturized systems for different application fields [12], because LTCC provides a convenient medium for fabricating laminated three-dimensional (3D) micro channel structures due to the easy and inexpensive fabrication process, low sintering temperature, and excellent mechanical, thermal, electrical, and chemical properties [13]. The LTCC microfluidic systems have been applied to many fields such as bioreactors, combustors, mixers, chemical reactors, heat exchangers, and gold nanoparticles generators [14]. The LTCC technology enables the fast and easy fabrication of microfluidic devices and systems. This can both reduce the cost of devices and shorten the development time [15].

Considering the above advantages of LTCC, a miniaturized XMFCD based on LTCC is proposed in this paper. Firstly, the design and fabrication of the LTCC XMFCD, which contains complex 3D micro channels, is illustrated, and the single and dual XMFCD samples are both produced and checked. Secondly, the mathematical model describing the flow characteristics of the XMFCD is established with the help of finite element analysis of the pressure distribution. Thirdly, the experiment setup, which is used to measure the mass flow of the XMFCD, is built. Fourthly, the pressure-flow characteristics of the single and dual XMFCDs are both measured to prove the proposed mathematical model, and the influence of different structural parameters on flow characteristics is discussed. Finally, the conclusion is outlined.

## 2. Design and Fabrication

The proposed XMFCD in this paper applies a labyrinth type microfluidic passage to realize ultra-high fluid resistance as mentioned in [9]. However, in [9], they used different implementation methods. The XMFCD in [9] is assembled by diffusion welding of several machined metal disks. However, the proposed XMFCD in this paper is produced by LTCC technology, which realizes miniaturization, low cost, and a short production cycle. The proposed XMFCD realizes ultra-high fluid resistance by complex 3D micro channels, which are composed of many chambers, grooves, and orifices in series. The chambers and grooves are embedded in two planes, and the orifices connect with the chambers in two planes.

The production process of the proposed XMFCD based on LTCC is shown in Figure 1. The LTCC technology and material system enable the easy creation of complex three-dimensional microfluidic

structures through structuring and assembly before the material is transformed into a rigid glass ceramic device. The starting point in LTCC technology is a green ceramic tape produced by a tape casting method, and various shapes (channels, cavities, vias, etc.) are cut in green LTCC tapes using a laser cutting and drilling machine (SK-MPL50, SANKE, Shanghai, China) in the first step. As shown in Figure 1, there are four shapes of green ceramic tapes (A, B, C, and D). The proposed XMFCD is composed of 14 tape layers (about 0.1 mm thickness each layer), which build the desired complex microfluidic passages. Layers 1, 2, 3, and 4 are A shape tapes which form a central inlet passage; Layer 5 is a B shape tape which hollows out 10 chamber-groove units; Layers 6, 7, 8, and 9 are C shape tapes which form 19 orifices; Layer 10 is a D shape tape which hollows out 10 chamber-groove units; and Layers 11, 12, 13, and 14 are A shape tapes which form a central outlet passage. In the third step, all the stacked LTCC tapes are laminated by a hydraulic press machine (6606-603-400, KISTLER, Beijing, China). Typically, the thermo-compression lamination process is performed at high pressure (up to 20 MPa), elevated temperature (up to 90 °C) for 5 to 30 min. After lamination, the LTCC module is co-fired according to a two-step thermal profile with a maximum temperature of 850 to 900 °C in a muffle furnace (SJL-200, CETC, Changsha, China) [16]. After the co-firing, the multi-layers are sintered into a hard piece. It should be noted that Figure 1 is a simplified schematic diagram for facilitating the disclosure of the manufacturing process which only includes one unit. In the actual processing, the 12 tape layers are all rectangular, which is easy to align. Each layer arrays 20 to 30 same-shape channel units (A, B, C, or D); therefore, 20 to 30 XMFCDs can be produced on one sintering board and then be cut to 20 to 30 independent circle XMFCDs, as shown in Figure 2a, by a laser cutting and drilling machine. This is why the LTCC production is efficient and cheap. The XMFCD made of 14 layers is called "single XMFCD" whose diameter is 9 mm and thickness is 1.4 mm. The mass of the single XMFCD is only 0.2 g, while the existing metal-based XMFCD with the same fluid resistance is at least tens of grams. In order to check the construction effect of the microfluidic channel, X-ray detection of the single XMFCD was performed, as shown in Figure 2b, which indicates that it matches the desired microfluidic channels. The formed complex 3D microfluidic structures of the proposed single XMFCD include 19 small orifices (diameter 0.1 mm), 20 thin grooves (width 0.2 mm, depth 0.1 mm) and 40 chambers (diameter 1 mm, depth 0.1 mm). Fluid enters at the center of the XMFCD and passes through a groove, which is tangential to a spin chamber. Then the fluid discharges through a small center hole into another chamber. This process repeats over and over to realize the ultra-high fluid resistance characteristic.

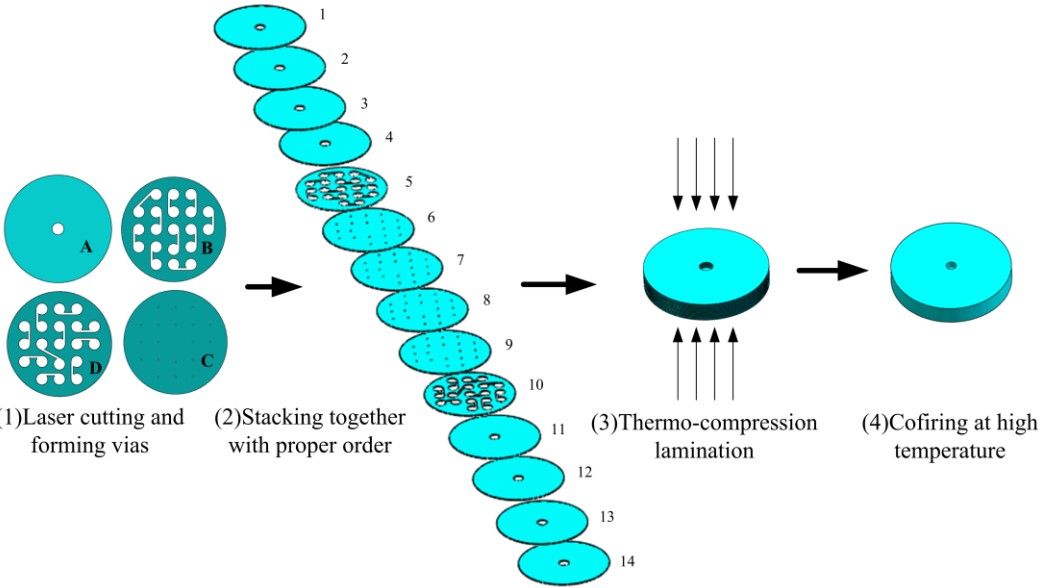

**Figure 1.** Fabrication process of the proposed xenon micro flow control device (XMFCD) based on low-temperature co-fired ceramics (LTCCs).

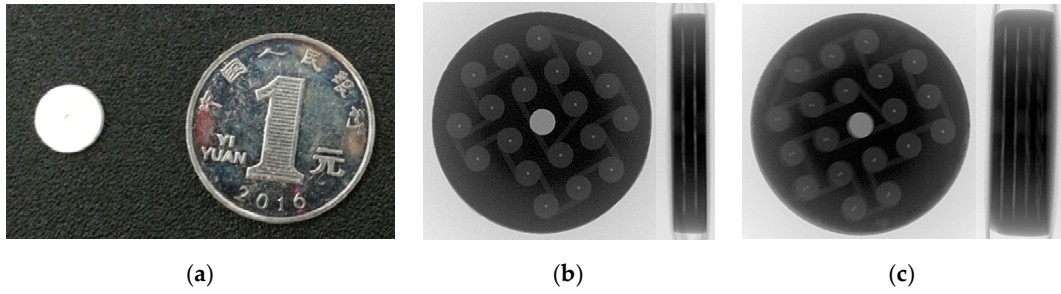

(a)　　　　　　　　　　　　(b)　　　　　　　　　　　　(c)

**Figure 2.** The proposed XMFCD samples and X-ray images. (**a**) Photograph of proposed XMFCD; (**b**) X-ray images of single XMFCD; (**c**) X-ray images of dual XMFCD.

In order to realize bigger fluid resistance, several units can be stacked together by LTCC technology. According to the fabrication process shown in Figure 1, dual XMFCD can be produced by 24 tape layers, which is an increase of 10 layers on the basis of the single XMFCD. The 15th to 24th layers of the dual XMFCD are the same as the Layer 5 to Layer 14 of the single XMFCD. A dual XMFCD was fabricated by LTCC, and its diameter and thickness are, respectively, 9 and 2.4 mm. Its mass is only 0.38 g, while the existing metal-based XMFCD with the same fluid resistance is at least tens of grams. The 3D microfluidic structure of the dual XMFCD includes 38 small orifices (diameter 0.1 mm), 40 thin grooves (width 0.2 mm, depth 0.1 mm) and 80 chambers (diameter 1 mm, depth 0.1 mm). The dual XMFCD is also detected by X-ray, as shown in Figure 2c, which indicates that it matches the designed fluid channel. Compared to the existing metal-based XMFCDs, LTCC-based XMFCDs are smaller, lighter, and cheaper.

## 3. Modelling and Simulations

For the proposed XMFCD, its mass flow characteristic is the most important performance parameter. In order to investigate the influence of structural parameters on the flow characteristics and guide the design of the proposed XMFCD, its mathematical model should be established. According to 3D microfluidic structures of the proposed XMFCD, the fluid model of the XFCD can be divided into two types of fluid units (Fluid Unit A and Fluid Unit B). Fluid Unit A contains 2 chambers, 1 groove, and 1 orifice, and Fluid Unit B contains 2 chambers and 1 groove. The single XMFCD consists of 19 Fluid Unit As and 1 Fluid Unit B in series, which includes 40 chambers, 19 orifices, and 20 grooves in total. The dual XMFCD consists of 38 Fluid Unit As and 2 Fluid Unit Bs in series which totally includes 80 chambers, 38 orifices, and 40 grooves. According to the actual working condition, the simulation block diagram of the proposed single and dual XMFCDs is given as shown in Figure 3. Therefore, the mathematical model of the proposed XMFCD is composed of 3 types of models: chamber, groove and orifice.

For the groove of the fluid unit shown in Figure 3, the mass flow through the groove can be treated as laminar flow, which can be described as [17]

$$Q_g = \frac{\pi D_g^4 \rho}{256 \mu R T_1 l_g} (P_1^2 - P_2^2) \tag{1}$$

where, $\rho$ and $\mu$ are, respectively, the density and viscosity of xenon gas; $R$ is the gas constant; $T_1$ is the absolute temperature of the gas in the groove; $D_g = \sqrt{4 w_g d_g / \pi}$ and $l_g$ are, respectively, the equivalent cross-sectional diameter and length of the groove; $w_g$ and $d_g$ are, respectively, the width and depth of the groove; and $P_1$, $P_2$ are the pressure of chambers at the upstream and downstream of the groove.

For the orifice of Fluid Unit A shown in Figure 3, the mass flow through the orifice can be expressed by [18]

$$Q_o = C_d A \sqrt{k} \frac{P_2}{\sqrt{R T_2}} f\left(\frac{P_3}{P_2}\right) \tag{2}$$

$$\mathrm{f}\left(\frac{P_3}{P_2}\right) = \begin{cases} \sqrt{\left(\frac{2}{k-1}\right)\left[\left(\frac{P_3}{P_2}\right)^{\frac{2}{k}} - \left(\frac{P_3}{P_2}\right)^{\frac{k+1}{k}}\right]} & \frac{P_3}{P_2} > P_{cr} \quad \text{subsonic flow} \\ \sqrt{\left(\frac{2}{k+1}\right)^{\frac{k+1}{k-1}}} & \frac{P_3}{P_2} \leq P_{cr} \quad \text{Supersonic flow} \end{cases} \tag{3}$$

$$P_{cr} = \left(\frac{2}{k+1}\right)^{\frac{k}{k-1}} \tag{4}$$

where, $C_d$ is the flow coefficient; $A = \pi D_o{}^2/4$ is the cross-sectional area; $D_o$ is the diameter of the orifice; $k$ is the gas adiabatic index; $T_2$ is the absolute temperature of the upstream gas of the orifice; $P_2$, $P_3$ are, respectively, the upstream and downstream pressure of the orifice; and $P_{cr}$ is the critical pressure ratio.

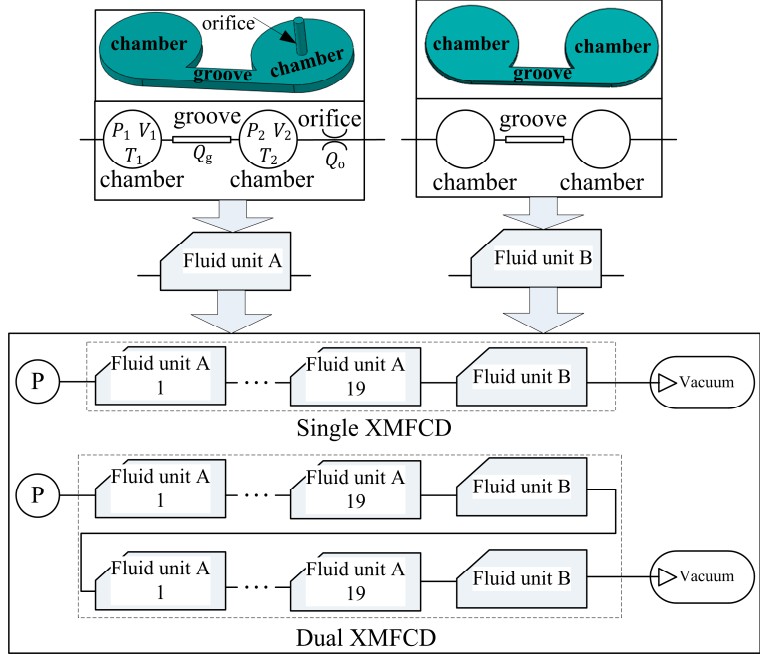

**Figure 3.** The simulation block diagram of the proposed XMFCD.

The flow characteristic model of the chamber can be expressed by [19] (for example, chamber $V_2$ shown in Figure 3)

$$\frac{dP_2}{dt} = \frac{R}{V_2}(T_1 Q_g - T_2 Q_o) \tag{5}$$

where, $V_2 = \pi D_c{}^2 d_g/4$ is the volume of the chamber, and $D_c$ is the diameter of the chamber.

The mass flow of the proposed XMFCD can be iteratively calculated by Equations (1)–(5) when its inlet pressure and outlet pressure are both known as the boundary conditions. However, the flow status through each orifice must be known in order to determine the subsonic flow equation or the supersonic flow equation in Equation (3) [20]. The finite element method by ANSYS is the most effective tool to do numerical analysis on flow and pressure characteristics of microchannels [21,22]. In this work, ANSYS is also applied to simulate the pressure distribution of the single XMFCD. According to its actual working condition in a space electric propulsion system, the inlet pressure and outlet pressure of the single XMFCD are respectively set to 0.15 MPa and 0 because the outlet is directly connected to a vacuum environment. Figure 4 is the pressure distribution map and each chamber's pressure is marked and plotted in Figure 5. Then the ratio of the outlet pressure and inlet pressure of each orifice is calculated and compared with the critical pressure ratio, $P_{cr}$. In the 40 chambers, only the ratio of the 39th chamber pressure and the 38th chamber pressure is smaller than $P_{cr}$ (for xenon, $k = 1.67$ and $P_{cr} = 0.4867$). Hence, only the flow of the last orifice (19th orifice) is supersonic

flow and the other 18 orifices are all subsonic flow. The proposed single and dual XMFCDs both have this feature because their outlets connects to a vacuum.

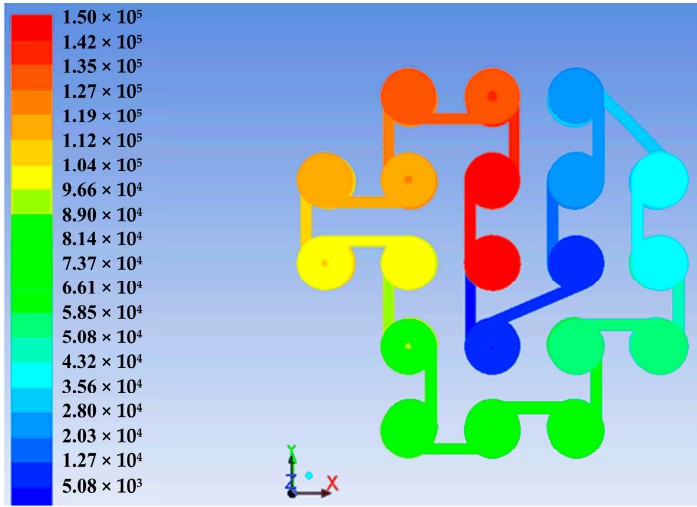

**Figure 4.** The pressure distribution map of single XMFCD.

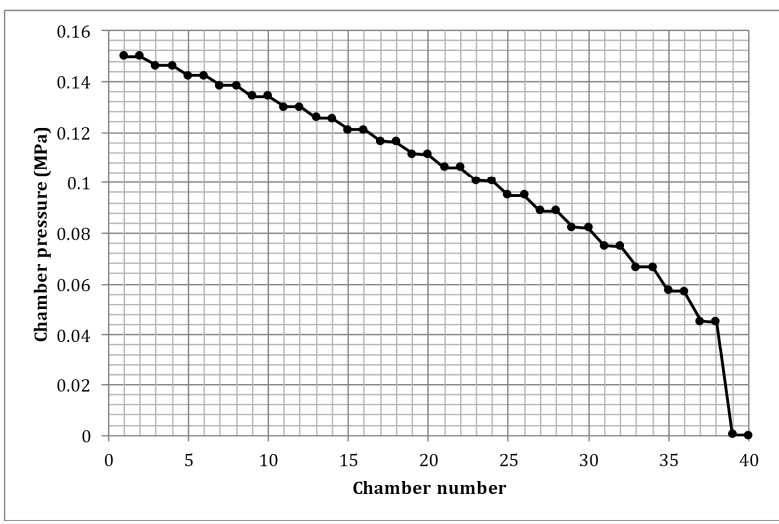

**Figure 5.** The chamber's pressure values of the single XMFCD.

## 4. Experimental Setup

Figure 6 presents the experimental setup, which is used to measure the mass flow of the proposed XMFCD. It was built with three manual on/off valves (SS-41GXS1, Swagelok, Solon, OH, USA), three ranges of high-precision mass flow meters (ALICAT SCIENTIFIC, 0–6, 0–20, and 0–100 sccm, Tucson, AZ, USA) and a dry scroll vacuum pump (IDP3A01, Agilent Technologies, Santa Clara, CA, USA). The LTCC XMFCD with a test connecter is installed between the high-precision flow meter and the dry scroll vacuum pump. The manual valves are used to control the on/off of the upstream xenon. The dry scroll vacuum pump vacuums the downstream of the LTCC XMFCD to simulate its actual operating conditions in an electric propulsion system. According to the flow range of the proposed LTCC XMFCD, the 0–20 sccm high-precision flow meter is chosen to measure the micro mass flow. A 5-L xenon tank with 5 MPa pressure supplies xenon for the experiment, which is not shown in Figure 6. The LTCC XMFCD's upstream xenon pressure, which is sampled by a high-precision pressure sensor (P30, 0–0.6 MPa, Wika, Frankfurt, Germany), is precisely adjusted by two pressure regulators (high

pressure: KPR1JWA422A20000RD, Swagelok, Solon, OH, USA; low pressure: KLF1FRA411A200000G, Swagelok, Solon, OH, USA) which are not shown in Figure 6.

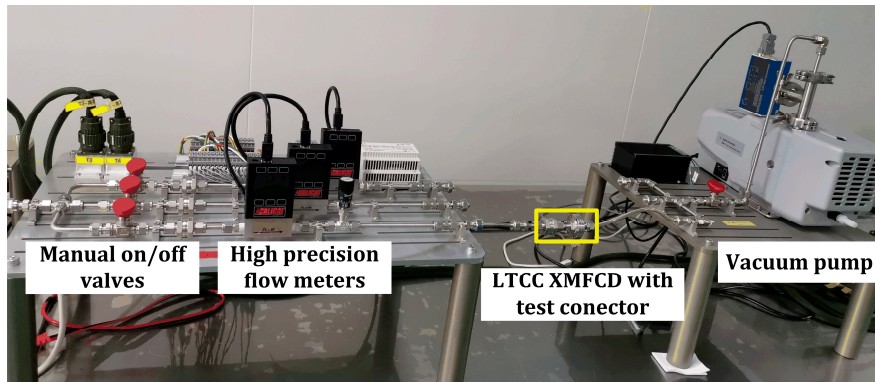

**Figure 6.** The experiment setup to measure mass flow of the proposed XMFCD.

This experiment setup is used to measure the steady flow of the proposed LTCC XMFCD. At the beginning of the experiment, the vacuum pump is opened and the desired upstream xenon pressure adjusted by two pressure regulators. Then when the value of the flow meter is displayed as 0, which proves the pipeline is already a vacuum, the manual on/off valve is opened and xenon gas is supplied to the LTCC XMFCD. Finally, after the flow meter is stable, the flow value is recorded.

## 5. Results and Discussions

### 5.1. Validation of the Mathematical Model

The flow characteristics of the single and dual XMFCDs are both simulated by the proposed mathematical model. The boundary conditions of the simulation are the same as the working conditions in an electric propulsion system. The upstream xenon pressure of the XMFCD is set to 0.1 to 0.2 MPa and the downstream pressure of the XMFCD is set to 0. The temperature is set to 20 °C. The structure parameters of the proposed XMFCD and the physical properties of the xenon are both listed in Table 1.

**Table 1.** The simulation parameters of the XMFCD.

| Parameters | Values |
|:---:|:---:|
| $\rho$ | 5.89 kg/m$^3$ |
| $\mu$ | $2.11 \times 10^{-5}$ Pa·s |
| $R$ | 63.29 J/(kg·K) |
| $k$ | 1.67 |
| $w_g$ | 0.2 mm |
| $d_g$ | 0.1 mm |
| $l_g$ | 1 mm |
| $C_d$ | 0.7 |
| $D_o$ | 0.1 mm |
| $D_c$ | 1 mm |

The flows at 11 pressure points between 0.1 to 0.2 MPa are calculated by the proposed mathematical model. In order to validate the mathematical model, the flows at the same pressures are tested. The measured and simulated mass flows of the single and dual XMFCDs are both listed in Table 2. Moreover, its errors are also calculated, which is shown in Table 2. The experimental and simulated flows of the single and dual XMFCDs are both plotted in Figure 7a,b, which show that the simulation flow results are in good agreement with the experiment results. According to Table 2, the error's absolute value of the measured and simulated mass flows of the single XMFCD is 0.00256 to 0.49 mg/s, which is 0.2% to

5.5% of the experimental results. Moreover, the error's absolute value of the measured and simulated mass flows of the dual XMFCD is 0.00318 to 0.03589 mg/s, which is 0.3% to 5.7% of the experimental results. Therefore, the mathematical model proposed in this paper is proven to be very effective for predicting the flow characteristics of the proposed XMFCD. Besides, the experimental results show that the mass flow of the proposed XMFCD is linearly proportional to the inlet pressure.

**Table 2.** The experimental and simulated mass flow of the proposed XMFCD.

| | Inlet Pressure (MPa) | Experimental Mass Flow (mg/s) | Simulated Mass Flow (mg/s) | Error (mg/s) |
|---|---|---|---|---|
| Single XMFCD | 0.1 | 0.880 | 0.92900 | −0.04900 |
| | 0.11 | 0.993 | 1.02488 | −0.03188 |
| | 0.12 | 1.095 | 1.12077 | −0.02577 |
| | 0.13 | 1.210 | 1.21667 | −0.00667 |
| | 0.14 | 1.310 | 1.31256 | −0.00256 |
| | 0.15 | 1.413 | 1.40847 | 0.00453 |
| | 0.16 | 1.520 | 1.50437 | 0.01563 |
| | 0.17 | 1.619 | 1.60027 | 0.01873 |
| | 0.18 | 1.719 | 1.69618 | 0.02282 |
| | 0.19 | 1.817 | 1.79209 | 0.02491 |
| | 0.20 | 1.910 | 1.88800 | 0.02200 |
| Dual XMFCD | 0.1 | 0.622 | 0.65789 | −0.03589 |
| | 0.11 | 0.702 | 0.72673 | −0.02473 |
| | 0.12 | 0.775 | 0.79558 | −0.02058 |
| | 0.13 | 0.856 | 0.86445 | −0.00845 |
| | 0.14 | 0.927 | 0.93331 | −0.00631 |
| | 0.15 | 0.999 | 1.00218 | −0.00318 |
| | 0.16 | 1.075 | 1.07106 | 0.00394 |
| | 0.17 | 1.145 | 1.13994 | 0.00506 |
| | 0.18 | 1.215 | 1.20882 | 0.00618 |
| | 0.19 | 1.285 | 1.27771 | 0.00729 |
| | 0.20 | 1.351 | 1.34659 | 0.00441 |

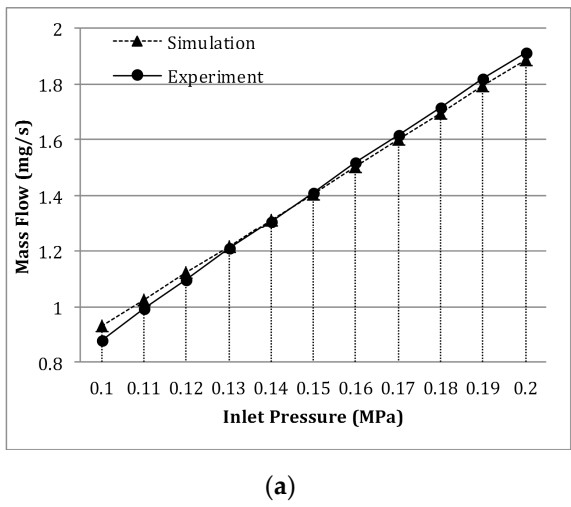
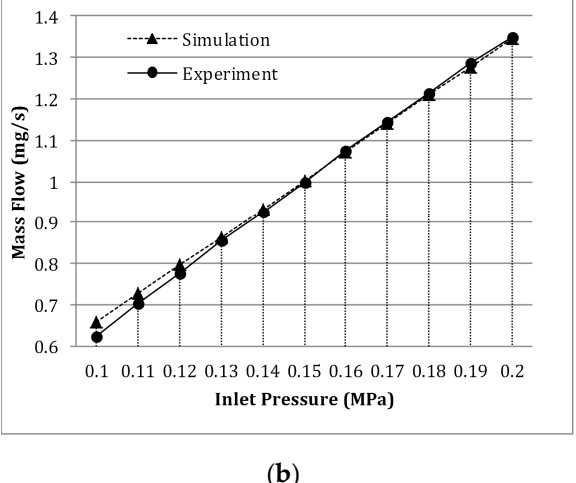

(**a**)         (**b**)

**Figure 7.** Comparison of the experimental and simulated mass flow of the proposed XMFCD. (**a**) Single XMFCD; (**b**) Dual XMFCD.

## 5.2. Influence of Structural Parameters

In order to realize the optimization design of the proposed XMFCD, the influence of structural parameters on flow characteristics should be investigated. According to Figure 3, the diameter of orifice $D_o$, and the width of groove $w_g$ are the parameters that are easy to adjust under volume limitation.

So the influence of these two parameters on the flow characteristic is focused on analysis using the mathematical model validated above. In the following simulations, the upstream xenon pressure of the XMFCDs is set to 0.15 MPa.

Firstly, the mass flow of the single and dual XMFCDs with different $D_o$ (40, 50, 60, 70, 80, 90, and 100 μm) and other parameters the same as shown in Table 1 are simulated. Moreover, the simulated mass flow results are shown in Table 3 and Figure 8a. According to the data analysis, two conclusions can be obtained as follows: (1) for XMFCDs with the same structural parameters, the mass flow $Q_1$ of the single XMFCD can be approximately described as 1.4 times of the mass flow $Q_2$ of the dual XMFCD, that proves the flow of N series orifices is $1/\sqrt{N}$ of a single orifice's flow, and (2) for the proposed XMFCDs, the mass flow is proportional to the square of the orifice diameter.

**Table 3.** The simulated mass flow of the XMFCD with different diameters of orifice ($D_o$) and widths of groove ($w_g$).

| Parameters | Value (μm) | Mass Flow $Q_1$ of Single XMFCD (mg/s) | Mass Flow $Q_2$ of Dual XMFCD (mg/s) | $Q_1/Q_2$ |
|---|---|---|---|---|
| $D_o$ | 40 | 0.23 | 0.165 | 1.393939 |
|  | 50 | 0.358 | 0.256 | 1.398438 |
|  | 60 | 0.514 | 0.368 | 1.396739 |
|  | 70 | 0.698 | 0.499 | 1.398798 |
|  | 80 | 0.908 | 0.649 | 1.399076 |
|  | 90 | 1.145 | 0.817 | 1.401469 |
|  | 100 | 1.41 | 1 | 1.41 |
| $w_g$ | 30 | 0.5976 | 0.3289 | 1.816966 |
|  | 50 | 1.021 | 0.637 | 1.602826 |
|  | 80 | 1.258 | 0.852 | 1.476526 |
|  | 100 | 1.321 | 0.914 | 1.445295 |
|  | 130 | 1.368 | 0.961 | 1.423517 |
|  | 150 | 1.385 | 0.978 | 1.416155 |
|  | 180 | 1.401 | 0.995 | 1.40804 |
|  | 200 | 1.41 | 1 | 1.41 |

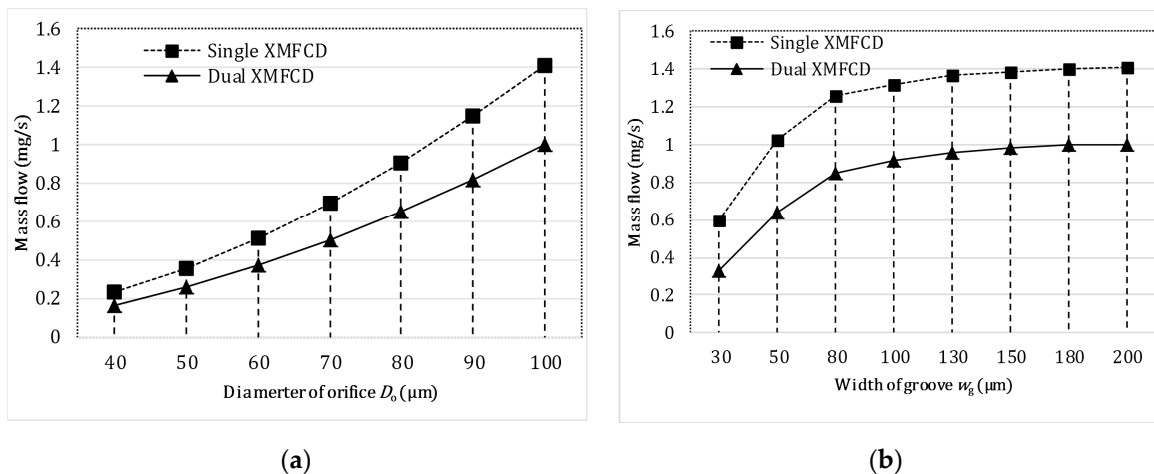

(**a**)　　　　　　　　　　　　　　　　　　　　　　　　(**b**)

**Figure 8.** The simulated flow of the XMFCD with different structural parameters. (**a**) Simulated flow with different $D_o$; (**b**) Simulated flow with different $w_g$.

Then, the mass flow of the single and dual XMFCDs with different $w_g$ (30, 50, 80, 100, 130, 150, 180, and 200 μm) and other parameters the same as shown in Table 1 are simulated. The simulated mass flow results are listed in Table 3 and drawn in Figure 8b. Referring to the simulated data, we can get conclusions as follows: (1) when $w_g \geq 80$ μm, the flow of the XMFCD is mainly determined by the orifices and the effect of the grooves on the flow is not obvious, and the mass flow in this case conforms

to the above flow formula of the series orifices; and (2) when $w_g < 80$ μm, the flow of the XMFCD decreases sharply as $w_g$ decreases because the throttling effect of the grooves becomes increasingly obvious. It is worth noting that when $w_g = 80$ μm, its equivalent diameter of the groove is about 100 μm which is as same as the diameter of the orifices. Therefore, in other words, only when the equivalent diameter of the groove is smaller than the diameter of the orifices, the impact of the groove on the XMFCD's flow will become significant.

## 6. Conclusions

This work proposed a miniaturized, easily processed, and inexpensive XMFCD, which is based on 3D microfluidic channels and fabricated by LTCC. The single XMFCD (diameter 9 mm, thickness 1.4 mm) and dual XMFCD (diameter 9 mm, thickness 2.4 mm) samples were both produced, and their 3D microfluidic structures were both detected by X-ray which proves the feasibility of this method. Moreover, other XMFCDs with more layers can be fabricated based on LTCC in order to obtain bigger fluid resistance, which can meet smaller flow requirements of space electric thrusters. The flow characteristic mathematical model and the simulation diagram validated by experiment results can be used to analyze and design other gas flow control devices based on microfluidic channels. The influence analysis of the structural parameters on flow characteristics reveals that (1) the diameter of the orifice is the most important parameter, whose square is proportional to the mass flow; (2) the mass flow of N series orifices is approximately $1/\sqrt{N}$ of a single orifice's flow, which can be used to estimate the flow of XMFCDs with different layers; and (3) only when the groove's equivalent diameter is smaller than the orifice diameter, the impact of the groove on the XMFCD's flow will become significant. So this work proposed a novel processing method for realizing the miniaturization of XMFCDs which can also be used in other micro flow control fields, and the mathematic model and simulation method lays a theoretical foundation for research on the flow characteristic of micro gas flow control devices.

**Author Contributions:** Conceptualization, C.-B.G. and Y.S.; Data curation, K.N. and H.-H.H.; Formal analysis, M.-J.Z. and K.N.; Investigation, H.-H.H. and Z.-L.W.; Methodology, C.-B.G.; Project administration, Y.S. and Z.-P.Y.; Software, M.-J.Z. and Z.-L.W.; Writing—original draft, C.-B.G.; Writing—review & editing, M.-J.Z. and Z.-P.Y.

**Funding:** This research was funded by National Natural Science Foundation of China, grant number 51805026.

**Conflicts of Interest:** The authors declare no conflict of interest.

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
