# Peer review of "Design, Simulation, and Experiment of an LTCC-Based Xenon Micro Flow Control Device for an Electric Propulsion System"

_processes, doi:10.3390/pr7110862_

Round 1

Reviewer 1 Report

see attached

Author Response

Dear reviewer,

Thank you for your recognition of this article. The following is the reply to your comments.

COMMENT 1: The authors claim there is interest in new low-cost, light-weight system for EP, which is true. But this is not the focus of their work, they never quantify or compare how this system cost/weight compares vs. other approaches. It is not necessary at this stage, but the authors should better highlight the need/interest in alternative approaches, and then justify that their approach has the potential to be light weight/low cost.  In other words:  Why do they expect this approach (LTCC) would be lower cost, lighter weight, and therefore more attractive than alternative approaches?

REPLY 1: this comment is very useful and we have added some description about the comparison between LTCC and the existing approaches. (1) we have described the disadvantages of the existing approaches in line 59/60 as “Besides, all above XMFCDs are made of metal, which results in big weight and difficulty to constructing microfluidic channel.”. (2) we have made further explanation of LTCC’s low cost in line 109-114 as “It should be noted that Figure 1 is a simplified schematic diagram for facilitating the disclosure of the manufacturing process which only includes one unit. In the actual processing, the 12 tape layers are all rectangle which is easy to align. Each layer arrays 20-30 same shape channel units (A, B, C or D), therefore 20-30 XMFCDs can be produced in one sintering board and then be cut to 20-30 independent circle XMFCDs as shown in Figure 2(a) by laser cutting and drilling machine. This is why the LTCC production is efficient and cheap.”. (3) we have added the mass data of the LTCC XMFCDs and compared to the existing approaches in line 116-117 as “The mass of single XMFCD is only 0.2 grams while the existing metal-based XMFCD with same fluid resistance is at least tens of grams.” and in line 134-136 as “Its mass is only 0.38 grams while the existing metal-based XMFCD with same fluid resistance is at least tens of grams.”. (4)we gives the comparison conclusion in line 139-140 as “Comparing to the existing metal-based XMFCDs, LTCC-based XMFCDs are smaller, lighter and cheaper.”.

COMMENT 2: Pg. 2, line 57/58  “With the large demand for low-cost and light-weight electric propulsion systems from microsatellites and commercial satellites, the above XMFCDs are no longer able to meet the requirements of the xenon feeding system.”   It is unclear how the authors can or have justified this statement.  Even if these are high-cost, heavy-weight feed systems, if they are the only ones available, EP for microsats and commercial sats must use them.  Your goal is therefore to create a feed system that is low cost and light weight, then?  Then I expect later in the manuscript you will compare your feed system cost and weight vs. other feed systems cost and weight??

REPLY 2 is the same as reply 1.

 COMMENT 3: Pg. 6, line 180:   Section 4 has same title as section 3.

REPLY 3: we have corrected this mistake in line 196 as “4. Experimental Setup”.

COMMENT 4: Pg 7, Fig 7, and all experimental data:   experimental data require a detailed error/uncertainty analysis and explanation, and error bars on figure data points. What is the difference between numerical prediction and experimental measurements?  It is never quantified.

REPLY 4: we have added Table 2 in line 241 which lists the error of the simulated data and experimental data. Besides, the error/uncertainty analysis and explanation was added in line 233-237 as “According to Table 2, the error’s absolute value of measured and simulated mass flow of single XMFCD is 0.00256mg/s-0.49mg/s which is 0.2%-5.5% of experimental results. And the error’s absolute value of measured and simulated mass flow of dual XMFCD is 0.00318mg/s-0.03589mg/s which is 0.3%-5.7% of experimental results.”.

COMMENT 5: What pressure is the diameter variation study run at?

REPLY 5: the pressure in the diameter variation was set in line 251-252 as “ In the following simulations, the upstream xenon pressure of the XMFCDs is set to 0.15MPa.”.

Reviewer 2 Report

This manuscript design a novel LTCC-based micro flow control device for electric propulsion system. As a new fabrication processing method, the description of real device in detail is required. Seems there is no information about the exact dimension of cofired samples, such as diameter of orifice, width of groove. Most of values in the manuscript are from simulation or given input numbers. For multilayer thick film processing, how about the feasibility and reliability of creating channels or vias with diameter less than 100 um? The alignment of small vias and deformation of fine vias/channel during the lamination could be issues. I suggest these information should be included and discussed. Microstructure images from SEM are needed to show the real dimension of vias and channels. The experiment result matching the simulation results should be based on the simulation initial/input consistent with experimental parameters. Without these information, the experimental validation of simulation and modeling will be not solid.  

Overall, this work was well conducted and the manuscript was well written.

Author Response

Dear reviewer,

Thank you for your recognition of this article. The following is the reply to your comments.

COMMENT 1: As a new fabrication processing method, the description of real device in detail is required.

REPLY 1: we have added the description of real device using in LTCC processing in line 97-108 which including laser cutting and drilling machine (SK-MPL50, SANKE, China), hydraulic press machine (6606-603-400, KISTLER, China) and a muffle furnace (SJL-200, CETC, China).

COMMENT 2: Seems there is no information about the exact dimension of cofired samples, such as diameter of orifice, width of groove. Most of values in the manuscript are from simulation or given input numbers.

REPLY 2: The dimension of cofired samples have been measured and record when their x-ray images were checked, but we have not marked it in the x-ray images. In the revision, we have added the description about the dimension in line 120-121 as “single XMFCD include 19 small orifices (diameter 0.1mm), 20 thin grooves (width 0.2mm, depth 0.1mm) and 40 chambers (diameter 1mm, depth 0.1mm)” and in line 136-138 as “dual XMFCD includes 38 small orifices (diameter 0.1mm), 40 thin grooves (width 0.2mm, depth 0.1mm) and 80 chambers (diameter 1mm, depth 0.1mm)”.

COMMENT 3: For multilayer thick film processing, how about the feasibility and reliability of creating channels or vias with diameter less than 100 um?

REPLY 3: the green ceramic tape is flexible which is easy to process and the laser drilling machine can precisely create channels or vias with diameter less than 100 um.

COMMENT 4: The alignment of small vias and deformation of fine vias/channel during the lamination could be issues.

REPLY 4: (1) We added an explanation to this issue in the revision in line 109-111 as “It should be noted that Figure 1 is a simplified schematic diagram for facilitating the disclosure of the manufacturing process which only includes one unit. In the actual processing, the 12 tape layers are all rectangle which is easy to align.”. (2) the deformation value of fine vias/channel during lamination and cofiring is basically fixed. When the channels in green ceramic tape are process, the amount of shrinkage is taken into account. so we can get the desired fluid channel after the lamination and cofiring.

COMMENT 5: Microstructure images from SEM are needed to show the real dimension of vias and channels.

REPLY 5: according to Reply 2, the dimension of cofired samples have been measured and record when their x-ray images were checked and we added the dimension in the revision.

COMMENT 6: The experiment result matching the simulation results should be based on the simulation initial/input consistent with experimental parameters. Without these information, the experimental validation of simulation and modeling will be not solid.  

REPLY 6: the real dimension of vias and channels is given and is consistent with the simulation initial/input.

Reviewer 3 Report

The paper "Design, Simulation and Experiment of LTCC based Xenon Micro Flow Control Device for Electric Propulsion System" is an interesting topic and a further application of ceramic multilayer technologies. However, I have some short questions:

Since I haven't read about metallization, what is the advantage of using LTCC? Why not using HTCC? You write of high temperature sintering in Fig.1, however, high temperature cofiring at "low temperature cofired ceramics?" What about the roughness of your channels? Does it influence the gas flow? Can you improve that with improved sintering profiles?

Author Response

Dear reviewer,

Thank you for your recognition of this article. The following is the reply to your comments.

COMMENT 1: Since I haven't read about metallization, what is the advantage of using LTCC? Why not using HTCC?

REPLY 1: the XMFCD is a pure structural device with complex microfluidic channels and is not a electronic device. So it do not need metallization. Besides, the controlled gas of the XMFCD is xenon which has good compatibility with ceramic, so LTCC is more suitable for this device.

COMMENT 2: You write of high temperature sintering in Fig.1, however, high temperature cofiring at "low temperature cofired ceramics?"

REPLY 2: this is a relative statement and 800~900°C is low temperature for sintering process.

COMMENT 3: What about the roughness of your channels? Does it influence the gas flow? Can you improve that with improved sintering profiles?

REPLY 3: the roughness of the channels is about Ra3.2. It does not influence the gas flow because the flow in this channel is not yet micro-flowing. The roughness of the channels is depended on the green ceramic tape.